# How do ConvNets Understand Image Intensity?

Jackson Kaunismaa[1] and Michael Guerzhoy[1,2]
[1]University of Toronto, [2]Li Ka Shing Knowledge Institute at St. Michael's Hospital
`jackson.kaunismaa@mail.utoronto.ca, guerzhoy@cs.toronto.edu`

## Abstract

Convolutional Neural Networks (ConvNets) usually rely on edge/shape information to classify images. Visualization methods developed over the last decade confirm that ConvNets rely on edge information. We investigate situations where the ConvNet needs to rely on image intensity in addition to shape. We show that the ConvNet relies on image intensity information using visualization.

## 1  Introduction

Convolutional Neural Networks (ConvNets) usually rely on edge/shape information to classify images. Shape is much more important than color in most situations (see e.g. (Reppa et al., 2020) for experiments on ImageNet). Both animals (Hubel & Wiesel, 1962) and ConvNets (Zeiler & Fergus, 2014) process images by detecting edges and corners in them. However, in some scenarios, color/intensity is important for object classification. Examples include medical imaging where the intensity of the "shadows" in the image may be important (Ezoe et al., 2002), classification of objects where color is an important cue (e.g. bird species (Welinder et al., 2010) classification), and scene recognition for landscape photos where color is more informative than shape (Gallagher et al., 2004).

In this paper, we explore how a ConvNet classifies images when color/image intensity is important. For simplicity, we focus on situations where *image intensity* is important. We generate synthetic images which can only be classified by paying attention to intensity, in addition to shape. We achieve this by making the class depend on intensity via a highly non-monotonic function. We visualize how the ConvNet classifies the images to show that the ConvNet uses image intensity as an important cue when "seeing" our synthetic inputs.

## 2  Task: Synthetic Data

We generate a dataset of greyscale images, each of which contains a single object of uniform intensity. We include additive noise at multiple scales. Classifying "dark" vs "light" is trivial. In order to make the task difficult, we make the target output be a non-monotonic function of the intensity of the object. See Fig. 2. We consider three classes, with, e.g., Class 0 corresponding to image intensity ranges of 0 to 30, 120 to 150, and 210 to 240.

## 3  How Does the ConvNet Understand Intensity?

### 3.1  Classification

We train several ConvNets (see Appendix 5.4 for the details) on 250k synthetic images. We achieve a correct classification rate of 98.2% with a large enough network on the test set (91.7% with a smaller network), with a base rate performance of 35.3%.

For the smaller network, the correct classification performance drops to 84.5% from 91.7% if the pixels of the image are permuted so that there are no shape cues in the image, indicating that the network relies on shape cues as well as image intensity cues.

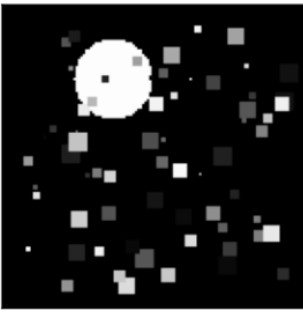 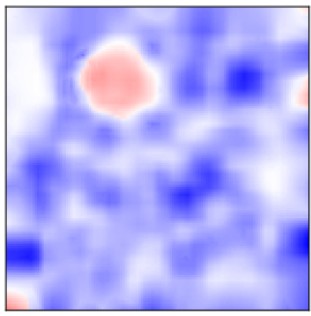 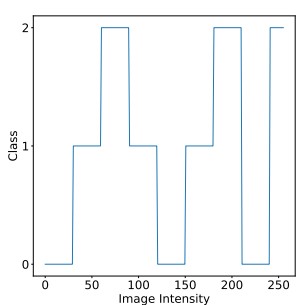

Figure 1: Left: Sample image from the synthetic dataset. The target is the class associated with the intensity of the grey circle, and the noise elements are the squares of various intensities and sizes. Right: Successful visualization of salient pixels. The interior of the circle is correctly identified as important.

Figure 2: The target function of the task. E.g., circles of intensity 30-60 are in Class 1, and circles of intensity 180-210 are in Class 2

## 3.2 VISUALIZATION

We visualize the pixels in the images that are important for the correct output using a method similar to Guided Backpropagation (Springenberg et al., 2015). On many test images, we see evidence that the network cares about the intensity in addition to the shape. We plot the average activation of a given convolutional channel across spatial dimensions to show that intermediate units in the network are highly dependent on intensity. Details are in the Section 5.3. A sample successful visualization is shown in Fig. 2.

## 4 CONCLUSION

We investigated how ConvNets understand images where the main cue is image intensity rather than shape. We demonstrated that it is possible to visualize the network to confirm that it uses intensity in addition to shape. Although shape is more important than color in most image data, our work could be useful in beginning to understand how ConvNets process image data where intensity and/or color is more important.

In this paper, we focused on scenarios where the target class is a highly non-monotonic function of the intensity of the input. Scenarios where the target class is a highly non-monotonic function of the 3D color are theoretically possible; however, among other issues, cameras are usually not calibrated with respect to color (Tedla et al., 2022).

Future work includes applying our methods and analysis to natural datasets, as well as empirically analyzing situations where the color, and not just the intensity, is important.

**URM Statement**: the authors qualify under the URM criteria.

*The authors thank Prof. Michael S. Brown for his insightful comments and suggestions.*

## 5 APPENDIX

### 5.1 PERFORMANCE

We trained several networks on both the task presented in section 2, as well as a modified, harder task, where the target is the same non-monotonic function of intensity, but the pixels in the input are permuted so that the network cannot rely on shape to identify the areas containing the relevant target intensity.

| Task | Network | Test Accuracy (%) |
|------|---------|-------------------|
| Unpermuted | Large | 98.2 |
| Unpermuted | Small | 91.7 |
| Permuted | Large | 97.6 |
| Permuted | Small | 84.5 |

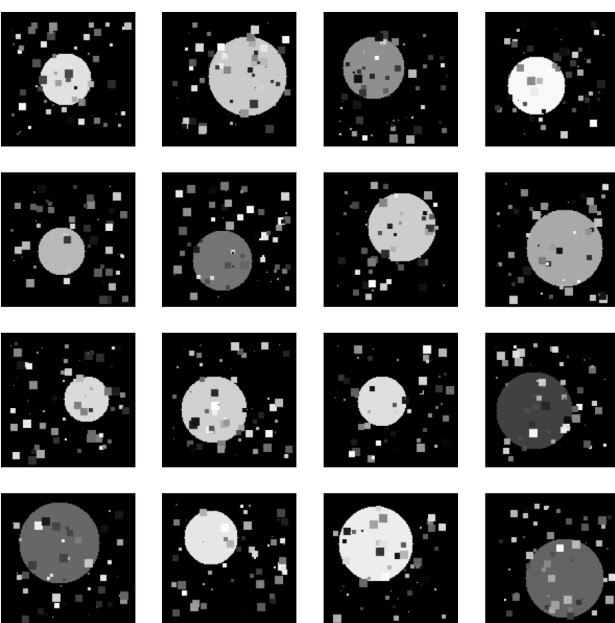

Figure 3: A selection of several examples from the unpermuted synthetic dataset.

### 5.2 INTENSITY ACTIVATION PLOT

In order to visualize how a given feature map $F$ responds to intensity, we display an "intensity activation plot." The activation at a given intensity is estimated by taking a large sample of synthetic images with circles at that intensity, and plotting the resulting average activation of feature map $F$ (Figure 5.2). We consider the activations after the ReLU, as they are essential for representing the "thresholding" behavior that intensity classifiers require. To visualize the variability in activation at a given intensity level, we plot each sampled (circle intensity, activation) pair, as a pale magenta dot. The average activation of feature map $F$ at a given intensity level is depicted with a red line. The reader can compare the red line with the regions corresponding to classes as specified by the non-monotonic target function to see how each channel "detects" a particular range of input intensities.

### 5.3 VISUALIZATION

We first visualized the network using Guided Backpropagation (**?**)[1]. Guided Backpropagation and similar methods for producing saliency maps are known to tend to emphasize edges in the image

---

[1] See (Joshi & Guerzhoy, 2017) Section IV.A for a presentation of Guided Backpropagation, which we recommend the reader familiarize themselves before reading this Section

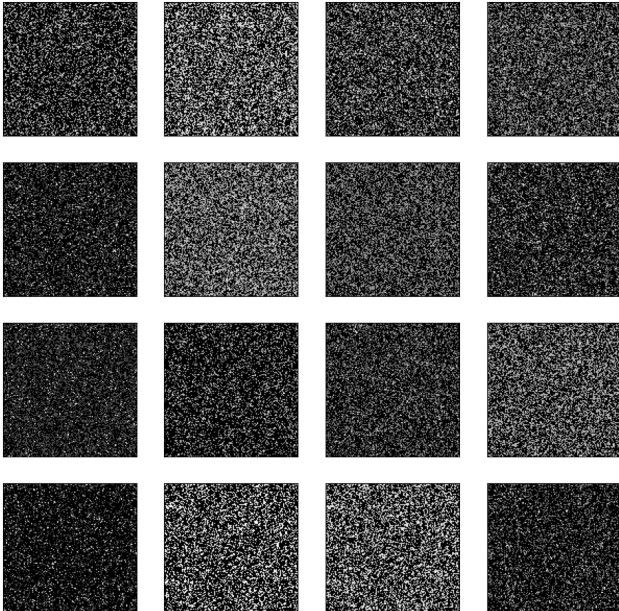

Figure 4: A selection of several examples from the modified synthetic dataset with permuted pixels.

regardless of whether they are actually important to the output (Adebayo et al., 2018). We modify Guided Backpropagation to allow us to demonstrate that our network does use pixels far away from edges.

Guided Backpropagation computes a saliency map indicating which locations in the input were important for the output using a modified version of the gradient of the output with respect to the input pixels. In our visualization, we compute a modified version of the gradient with respect to a local linear approximation of image patches, allowing as to visualize which *patches* in the images are important for the output.

We run Principal Component Analysis on a training set of image patches from our dataset. We then consider the quantity $f(I + \alpha p)$, where $I$ is the image, $p$ is a learned principal component of a patch, padded with 0's to be the same size as the image, and $f$ is our network. The quantity $\partial f(I+\alpha p)/\partial \alpha$ is then analogous to the gradient of the output with respect to the input. We modify the computation of $\partial f(I + \alpha p)/\partial \alpha$ analogously to Guided Backpropagation, removing the contributions of paths in the network that contain negative edges.

To compute the importance of a particular pixel in the image, we take the maximum over multiple scales of the modified $\partial f(I + \alpha p)/\partial \alpha$ for principal components $p$ for each pixel, and then linearly interpolate those quantities to obtain the saliency map.

## 5.4 MODEL ARCHITECTURE

For ease of visualization and interpretation, we considered both a "small" ConvNet and a "large" ConvNet consisting of the following layers:

| Layer | In Channels | Out Channels | Stride |
|-------|-------------|--------------|--------|
| 1 | 1 | 16 | 1 |
| 2 | 16 | 16 | 1 |
| 3 | 16 | 32 | 1 |
| 4 | 32 | 32 | 1 |

Table 1: Architecture of large network.

| Layer | In Channels | Out Channels | Stride |
|-------|-------------|--------------|--------|
| 1 | 1 | 2 | 4 |
| 2 | 2 | 2 | 1 |
| 3 | 2 | 6 | 4 |
| 4 | 6 | 6 | 1 |

Table 2: Architecture of small network.

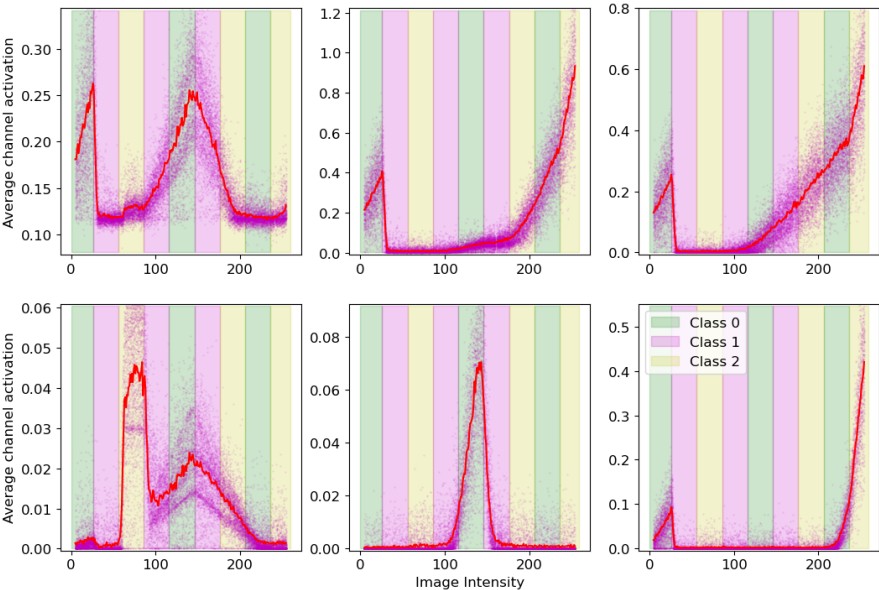

Figure 5: Visualization of an intermediate convolutional layer (specifically, the 5th layer of the small network). Each plot corresponds to a specific channel in that layer, plotting the average feature activation vs. the intensity of the circle in the input. We observe that each channel in the intermediate "detects" a particular range of input intensities that corresponds to parts of the non-monotonic target function.

The convolutional layers are followed by a single fully-connected layer with 3 output logits in each network, and all convolutions are with 3x3 kernels. Each layer in the network uses Batch Normalization and ReLU non-linearities.

Network hyperparameters were tuned with a random search over learning rate, initialization variance, and weight decay. They were trained using the Adam optimizer and the cross-entropy loss on $250,000$ samples from the synthetic dataset. We emphasize that the goal of this paper was not to obtain high performance on our specific dataset, but rather to analyze how convolutional networks solve the task.

## 5.5 SYNTHETIC DATASET GENERATION

The images in the synthetic dataset are generated with the following steps:

1. Initialize image $I$ as an $S$ x $S$ array of zeros.

2. Sample a radius uniformly at random from $r_{min}$ to $r_{max}$, and center location in pixel coordinates from $[r_{max}, r_{max} + 1, ..., S - r_{max}]$. Doing so guarantees that circles are always fully within the image.

3. Pick a random intensity for the circle and set the corresponding pixels in $I$ to that value.

4. Sample uniformly at random a number of noise elements to be added from $[n_{min}, n_{min} + 1, ..., n_{max}]$. For each noise element (pixel coordinate, size, intensity) tuples, corresponding to the square noise elements. Pixel coordinates range from 0 to $S$, size ranges from $w_{min}$ to $w_{max}$, and intensity ranges from 0 to 255. Doing so may overwrite the circle pixels generated in step 2.

5. (optional) If desired, apply a random permutation to the pixels in $I$.

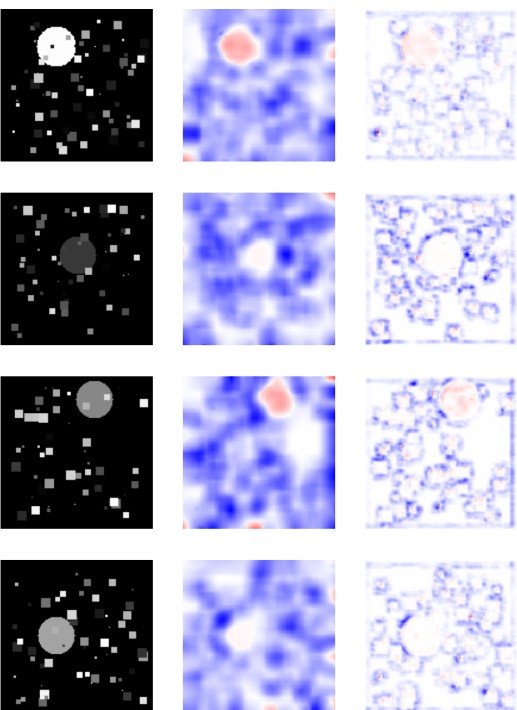

Figure 6: Selected examples of saliency maps. Column 1: image from dataset. Column 2: Saliency map computed using our method. Column 3: Guided Backpropagation.

For all experiments, we use $S = 128$, $(r_{min}, r_{max}) = (16, 42)$, $(n_{min}, n_{max}) = (50, 70)$, $(w_{min}, w_{max}) = (1, 9)$. To make results repeatable, we set the random seed of the PRNG.

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
