# OpenReview forum: "How do ConvNets Understand Image Intensity?"
_ICLR.cc/2023/TinyPapers — Submitted to Tiny Papers @ ICLR 2023_

### Official Review · Reviewer_N4dY · 2023-03-31

**Confidence:** 3

**Summary Of Contributions:**

This paper investigates situations where the ConvNet needs to rely on color rather than shape.

**Rating:**

Clear, Correct, and Reproducible (CCR): a submission which meets the reviewing criteria

**Strengths And Weaknesses:**

Strengths:
i) Various experiments have been conducted.
ii) This work is a first attempt for understanding how ConvNets process image data where color is more important.

Weaknesses:
i) The authors have not exploited a large network.
ii) References in text.

**Suggested Changes:**

Please correct the way references are shown in text. Use \citep.

---

> ### Author Response · Authors · 2023-05-31
> **Thank you for your comments**
>
>
> Thank you for the review and the suggestions.
>
> We have now included the results for all of the experiments, and fixed the typographical issues that you pointed out.

---

### Official Review · Reviewer_BZGb · 2023-04-01

**Confidence:** 5

**Summary Of Contributions:**

The paper investigates which information convolutional neural networks pay attention to in a generic image classfication task. Some experiments are designed and results are presented. The results of intermediate layers of the neural networks are shown to back the claim that CNNs look at color/brightness of pixels along with edge/shape of objects.

**Rating:**

Great Start (GS): a submission which meets some of the reviewing criteria but has room for improvement

**Strengths And Weaknesses:**

Strengths:
1. The topic discussed in the paper is interesting and the paper has a good flow.

Weaknesses:
1. "For simplicity, we focus on how ConvNets understand brightness, rather than 3-channel color." This seems like an alarming deviation from the title: "How Do ConvNets Understand Color?"
2. Following 1, brightness and color seem to be used interchangeably
3. The experiments seem to be designed such that the brightness is the defining factor of discrimination. It seems natural that the neural network picked up on that.

**Suggested Changes:**

1. "... with a base rate performance of 35.3%". How is this calculated? Random assignment of classes?

---

> ### Author Response · Authors · 2023-05-31
> **Thank you for your comments**
>
> Thank you for the thoughtful review.
>
> We agree with your concerns about the original version of the paper. We adjusted the terminology to always say “image intensity” when that is what we mean.
>
>
> “The experiments seem to be designed such that the brightness is the defining factor of discrimination. It seems natural that the neural network picked up on that.”
>
> We agree, but we would like to point out our experiments on the “permuted” and “unpermuted” images. On “permuted” images, where intensity is the only cue, the networks seem to average over pixels to get rid of some of the noise, but to otherwise only use intensity. On the other hand, in experiments where blobs of a constant color are present, like you say, shape and intensity cues are both used.
>
> “"... with a base rate performance of 35.3%". How is this calculated? Random assignment of classes?”
> We define the base rate of the classifier consistently with e.g. Shalizi https://www.stat.cmu.edu/~cshalizi/dm/19/lectures/23/lecture-23.html as the correct classification rate when the plurality class is picked every time as the answer.
>
> Jack & Michael

---

### Official Review · Reviewer_KRsj · 2023-04-02

**Confidence:** 2

**Summary Of Contributions:**

The paper uses synthetic with varying brightness and visualizes how Convent uses color.

**Rating:**

Clear, Correct, and Reproducible (CCR): a submission which meets the reviewing criteria

**Strengths And Weaknesses:**

The paper is concise and reads clearly. The investigation is necessary and an important one. The experiment setup introduced in this paper is a good starting point and can be used as a baseline for further investigations. The experiment could be extended to standard datasets.

The processing of images (feature processing) can affect ConvNet’s ability to identify cue from colors, adding a section on how exactly images were processed would make this reproducible.

I am not an expert in the field, is there a reason why the color identifying shape was a circle and the noise were squares? Also, I’d think if other variables are kept constant/randomized, ConvNets would identify color (brightness) as the “cue”. However, in any real world setting, shape and edges and color features will always be present  together in dataset. A natural extension of this experiment could be to concatenate color based features separately to improve ConvNets ability to take color as “cue”

**Suggested Changes:**

Change dataset where shape, edges and color are interacting together and measure ConvNets ability to use color to distinguish between similar images. Example : green grapes vs black grapes. Water wave forms vs sand wave forms etc.

---

> ### Author Response · Authors · 2023-05-31
> **Thank you for your comments**
>
> Thank you for the thoughtful review, and for your suggestions.
>
> You are correct that the choice to use circles/squares is arbitrary.
>
> “I am not an expert in the field, is there a reason why the color identifying shape was a circle and the noise were squares? Also, I’d think if other variables are kept constant/randomized, ConvNets would identify color (brightness) as the “cue”. However, in any real world setting, shape and edges and color features will always be present together in dataset.”
>
> We believe this is addressed to some extent by our experiments on the “permuted” and “unpermuted” images. On “permuted” images, where intensity is the only cue, the networks seem to average over pixels to get rid of some of the noise; but otherwise they use only intensity. On the other hand, in experiments where blobs of a constant color are present, like you say, shape and intensity cues are both used.
>
> We agree that working with natural images is an obvious next step, and we will be working on that.
>
> Jack & Michael

---

### Author Response · Authors · 2023-05-31
**Opt in to archive**

We would like to opt in into archiving the paper.

We thank the reviewers for their thoughtful reviews.

Jack & Michael

---

### Comment · Area_Chair_7YDb · 2023-06-02
**Archival**

This work meets the threshold for archival, contents the URM statement and is deanonymized

---

### Meta-Review · Area_Chair_7YDb · 2023-04-05

**Recommendation:** Invite to archive
**Confidence:** 4

**Metareview:**

The paper explores how CNNs understand color. Some experiments are designed to answer this question. The authors claim that CNNs relies on color by showing visualizations of intermediate layers.

**Summary:**

The reviewers recognize the problem the paper addresses and they express some concerns

**Reason For Not Giving A Higher Recommendation:**

Reviewers express some reproducibility concerns ("adding a section on how exactly images were processed would make this reproducible.") and significant deviation from the problem statement ("For simplicity, we focus on how ConvNets understand brightness, rather than 3-channel color." This seems like an alarming deviation from the title: "How Do ConvNets Understand Color?"). Ultimately none of the reviewers champion the paper so the chair requests the authors to address the concerns in the reviews.

**Reason For Not Giving A Lower Recommendation:**

The reviewers agree that while there are some concerns the topic presented has some potential.

---

> ### Author Response · Authors · 2023-05-31
> **Thank you for the review**
>
> We have addressed the reviewers' concerns. Please see our answer below.
>
> Thank you for helping make our paper better.
>
> Jack & Michael

---

### Decision · Program_Chairs · 2023-04-07

Invite to archive

---

> ### Author Response · Authors · 2023-05-31
> **Opt in to archive**
>
> We would like to opt-in into archiving the paper.
>
> We thank the reviewers for their thoughtful reviews.
>
> Jack & Michael